# Pathogenesis of Human Adenomyosis: Current Understanding and Its Association with Infertility

**DOI:** 10.3390/jcm11144057

**Published:** 2022-07-13

**Authors:** Khaleque N. Khan, Akira Fujishita, Taisuke Mori

**Affiliations:** 1Department of Obstetrics and Gynecology, Graduate School of Medical Science, Kyoto Prefectural University of Medicine, Kyoto 602-8566, Japan; moriman@koto.kpu-m.ac.jp; 2Department of Gynecology, Saiseikai Nagasaki Hospital, Nagasaki 850-0003, Japan; fujishita@nsaisei.or.jp

**Keywords:** adenomyosis, pathogenesis, epithelial-mesenchymal transition, HGF, estrogen, Müllerian remnants, stem/progenitor cells, somatic mutation, microRNAs, classification, infertility

## Abstract

The aim of this review article was to summarize our current understanding on the etiologies and pathogenesis of human adenomyosis and to clarify the relative association between adenomyosis and infertility. The exact pathogenesis of adenomyosis is still elusive. Among different reported concepts, direction invagination of gland cells from the basalis endometrium deep into myometrium is the most widely accepted opinion on the development of adenomyosis. According to this concept, endometrial epithelial cells and changed fibroblasts, abnormally found in the myometrium in response to repeated tissue injury and/or disruption at the endometrium-myometrium interface (EMI), elicit hyperplasia and hypertrophy of the surrounding smooth muscle cells. In this review, a comprehensive review was performed with a literature search using PubMed for all publications in English and Japanese (abstract in English), related to adenomyosis and infertility, from inception to April 2021. As an estrogen-regulated factor, hepatocyte growth factor (HGF) exhibits multiple functions in endometriosis, a disease commonly believed to arise from the functionalis endometrium. As a mechanistic basis of gland invagination, we investigated the role of HGF, either alone or in combination with estrogen, in the occurrence of epithelial-mesenchymal transition (EMT) in adenomyosis. Aside from microtrauma at the EMI, metaplasia of displaced Müllerian remnants, differentiation of endometrial stem/progenitor cells within the myometrium and somatic mutation of some target genes have been put forward to explain how adenomyosis develops. In addition, the possible role of microRNAs in adenomyosis is also discussed. Besides our knowledge on the conventional classification (focal and diffuse), two recently proposed classifications (intrinsic and extrinsic) of adenomyosis and the biological differences between them have been described. Although the mechanistic basis is unclear, the influence of adenomyosis on fertility outcome is important, especially considering the recent tendency to delay pregnancy among women. Besides other proposed mechanisms, a recent transmission election microscopic (TEM) study indicated that microvilli damage and an axonemal alteration in the apical endometria of human adenomyosis, in response to endometrial inflammation, may be involved in negative fertility outcomes. We present a critical analysis of the literature data concerning the mechanistic basis of infertility in women with adenomyosis and its impact on fertility outcome.

## 1. Introduction

Adenomyosis is an estrogen-dependent chronic inflammatory condition and is characterized by the presence of endometrial glands and stroma within the myometrium causing enlargement of the uterus as a result of reactive hyperplastic and/or hypertrophic change of the surrounding myometrium [1,2]. Adenomyosis was first described by pathologist Carl von Rokitansky as endometriosis interna in 1860 [3] and subsequently recognized as an elusive disease by gynecologist Ludwig Emage in 1962 [4]. The terminology of Rokitansky was later supported by Cullen after 50 years [5]. The term “adenomyosis uteri” was first designated by Frankl in 1925 and the current definition of adenomyosis that is widely used today was first introduced by Bird et al. in 1972 [1,2]. Two fundamental differences between endometriosis and adenomyosis are thought to be the origin of endometrium (functionalis or basalis) and anatomic site of the lesions (outside or inside of the uterus, respectively) [1,2]. Endometriosis and adenomyosis are closely related diseases with variable coexistence rates depending on the endometriosis phenotype involved [6,7,8]. Endometriosis and adenomyosis share a number of features, in terms of symptomatology, histology, and molecular alterations [9,10,11]; albeit, there are several differences in their pathogenesis and pathogenic mediators [12].

Several theories supporting the pathogenesis of adenomyosis have been advocated since the etiology of this condition has remained unclear for more than a half-century [13]. Among them, the two most common and widely accepted theories that have been postulated are invagination of the basalis endometrium into the myometrium and de novo origin from the metaplasia of embryonic Müllerian remnants or of endometrial stem/progenitor cells with the myometrium [6,14,15,16]. A panel of mechanisms has been reported, indicating tissue damage or injury at the endometrial-myometrial interface (EMI), leading to inflammation, local estrogen production and development of adenomyosis [13,14,15,16,17,18]. 

Adenomyosis commonly occurs during the fourth and fifth decades of life and after the completion of childbearing activity. However, recent imaging modalities, such as trans-vaginal ultrasonography and magnetic resonance imaging (MRI), have indicated that adenomyosis may occur in women of younger ages [1,19]. Women with adenomyosis suffer from different problems, such as decreased quality of life as a result of severe painful symptoms/abnormal uterine bleeding, and progressive disease may result in a state of subfertility/infertility demanding proper treatment [20,21]. 

Progress on the understanding of epidemiology, etiology and the pathogenesis of adenomyosis is poorly described, compared to other benign reproduction diseases. This might be due to long dependence on histopathologic examination of uterine specimens after hysterectomy for disease diagnosis and lack of reliable pre-operative diagnosis. Adenomyosis may cause a negative impact on reproductive outcomes and other health outcomes, including obstetrical complications. Despite its prevalence and the severity of its symptoms, little information is available on the etiology/pathogenesis of adenomyosis and our knowledge is insufficient on the factors related to negative fertility outcome in women with adenomyosis. In this review article, we aimed to summarize our current understanding on the different etiologies/pathogenic theories in adenomyosis and recent information on factors that might be associated with infertility. In this article, a comprehensive review was performed with a literature search using PubMed for all publications in English and Japanese (abstract in English), related to adenomyosis and infertility, from inception to April 2021. 

## 2. Epidemiology

The incidence rate of adenomyosis is widely variable. Histological examination of hysterectomy specimens revealed that the prevalence of adenomyosis varies between 5% and 70%. This variation may be due to the difference in diagnostic criteria used and the techniques used to procure myometrial samples [22,23,24]. A separate study with data over the last 50 years demonstrated that the estimated prevalence of adenomyosis among consecutive hysterectomy patients ranged from 8.8 to 61.5% [13]. All these studies have a similar opinion that lack of standardized histopathologic criteria for diagnosis, and the variable number of histologic tissue samples evaluated per hysterectomy, may be attributed to this wide range in incidence rate. A variable prevalence rate has been reported in other studies based on histopathologic criteria. A prevalence of 10% is reported with the diagnostic criteria of ≥5 mm distance between endometria glands and endo-myometrial junction and the presence of myometrial hyperplasia [24]. In another report, the prevalence was higher at 18% with an endometrial gland depth of only ≥1 mm and no myometrial hyperplasia [25]. Based on these findings, it has been suggested that nearly half of adenomyosis present in extirpated uteri remains undiagnosed [25,26]. 

Compared to white women, a greater prevalence of surgically confirmed adenomyosis among Latinas was reported by a large cohort study of over 80,000 female teachers in California [26]. In this study, a small sample size of non-white women impeded the evaluation of the association between black/Asian women and adenomyosis risk. In contrast, black women were more likely to have a pathologic finding of adenomyosis and leiomyomas than Hispanic women, as confirmed by a study of women undergoing hysterectomy in New York [27]. In a rare population-based study, the estimated prevalence of adenomyosis in Northeast Italy is reported to be 0.18%, much lower than that of endometriosis (1.14%) [28]. 

The risk profiles of adenomyosis are different from that of endometriosis, in which greater parity is a protective factor, while early age at menarche and short menstrual cycle are considered as promoting factors [29]. A different report claimed that risk factors associated with adenomyosis include early menarche, short menstrual cycle, increased body mass index, and a history of depression [26,27]. While parity is considered to be a risk factor for adenomyosis, smoking is shown to be a protective factor, compared with women who had never smoked [23,30,31]. Some other studies showed either higher rates of smoking in women with adenomyosis [32] or no association between them [33]. Two studies with lactating mothers found that breastfeeding is associated with the absence of ovulatory cycles and estrogen-deficiency [33,34]. Therefore, it is possible that among parous women, breastfeeding could be associated with a decreased risk of adenomyosis. A line of evidence suggests that a history of D and C procedures is a risk factor for adenomyosis [32,35,36,37]. The D and C procedures, and other uterine surgeries, such as Caesarian section and/or hysteroscopic procedure, may disrupt the EMI and facilitate the cellular migration/invasion, implantation, shedding, and development of endometrial colonies with the myometrium. These biological events of tissue injury/damage at EMI may clarify the higher risk of adenomyosis. The uninterrupted menstrual cycle and degradation of functionalis endometria until a later age of the women eliciting constant tissue stress reaction and/or tissue damage at EMI could be another added contributory factor in the development of adenomyosis.

A controversial issue still remains regarding the relationship between adenomyosis and estrogen exposure. Information on this issue is diverse. While one study failed to find a link between the risk of adenomyosis and use of oral contraceptives [32,33], another study reported increased rates of adenomyosis in patients using oral contraceptives [24,25]. Further investigation is required to confirm this information, because oral contraceptives are commonly used for symptoms of adenomyosis. Women given tamoxifen to treat breast cancer showed a higher prevalence of adenomyosis [1,38,39,40]. As a non-steroidal agent, while tamoxifen exhibits anti-estrogenic effects on breast tissue, it has a pro-estrogenic effect on the endometrium, thereby facilitating cell proliferation, migration and development of adenomyosis [41]. In addition to the risk factors mentioned above, some designated genetic and epigenetic alterations have also been detected in adenomyosis [42,43,44,45].

## 3. Biology of Endometrium and Myometrium

Endometriosis is commonly believed to originate from functionalis endometrium and adenomyosis from basalis endometrium. Menstrual shedding occurs from breakdown of functionalis endometrium, whereas underlying basalis endometrium remains intact during menstruation. A proportion of this degraded menstrual effluent goes into the pelvis through the patent Fallopian tubes, comes in contact with any of several anatomical sites (peritoneum, ovary, extra-genital sites) and results in the development of endometriosis (endometriosis externa). The viable residual basalis endometrium results in the development of adenomyosis (endometriosis interna) in response to repeated tissue injury or trauma at the endometrial-myometrial junction. However, our understanding on the biological differences between these two layers of endometrium and between inner and outer myometrium in women with adenomyosis is poor.

Full thickness (from the endometrium to the myometrium) biopsy specimens after hysterectomy from women with and without adenomyosis are needed to address this issue. We can distinguish basalis from functionalis endometria based on the criteria as described previously [17]: (i) constant appearance of basalis throughout the menstrual cycle, (ii) glands of the basalis layer appear weakly proliferative, (iii) basalis cells lack secretory features and stroma are spindled and non-decidualized. As an additional criterion, we can also define basalis endometrium as 1-mm area from the endo-myometrial junction, as previously reported by Japanese histopathologist Setoguchi in 1979 [46]. Considering these criteria to demarcate basalis from functional endometria, a case-controlled study using immunohistochemical analysis was performed which demonstrated that expressions of estrogen receptor (ER) and progesterone receptor (PR) were significantly lower in the basalis endometria than in the functionalis endometria during the secretory phase and the menstrual phase [5,17]. In contrast, a similar pattern of ER and PR expression was found in the functionalis and basalis endometria during the proliferative phase. This was equally observed in women with and without adenomyosis. In addition, Ki-67 index was lower and TUNEL-positive cells and activated Caspase-3-imunoreacitve cells were significantly lower in the basalis endometria, than those in the functionalis endometria [17]. These findings indicate that the two components of endometrium (functionalis and basalis) are biologically different and dictate a possible mechanistic basis of tissue degradation of the functionalis endometria and residual intact basalis endometria during menstruation. The surviving basalis endometria may consequently suffer a constant tissue stress reaction in response to cyclic breakdown of functionalis endometria. The fate of intact basalis endometria in the development of adenomyosis has been proposed [1,2]. The shedding of functionalis endometrium and intact basalis endometrium during the menstrual phase, and expression patterns of ER/PR/Ki-67/TUNEL positive cells in these two layers of endometrium, are shown in Figure 1.

A fair knowledge on the inner myometrium that lies in closed apposition with basalis endometrium, without any intervening basement membrane, is important to better understand the mechanistic basis of adenomyosis. Structurally, myometrium is composed of two layers: an outer longitudinal layer (outer myometrium, OM) and inner circular layer of smooth muscle cells (SMCs) (inner myometrium, IM) and these layers are supported by stromal and vascular tissue [47,48,49]. Since endometrial basalis is in direct contact with the inner myometrium, any biological or surgical insult may facilitate cross-communication of cells between these two layers and any damage at EMI may cause a pathological process to ensue. The sub-endometrial layer of the myometrium (IM, also called junctional zone) can be identified as T2-low signal intensity by magnetic resonance image (MRI) [50]. While MRI displays high-signal intensity for endometrium, it shows medium-signal intensity for outer myometrium on T2-weighted image [48,49,50]. This may be explained by developmental difference. IM arises from the Müllerian ducts, as does the endometrium, whereas OM is mesenchymal in origin [49]. This embryological relevance between endometrium and IM, as well as lacking of basement membrane in between, may further explain a possible involvement of these two layers in the development of adenomyosis. 

Histological studies reported that the density and arrangement of myocytes (SMCs) differ in IM and OM [49,51,52]. The IM exhibits irregular, densely packed, circular muscle fibers with increased vascularity, whereas OM shows regular and predominantly longitudinal SMC bundles with water content [50,51]. This may explain why IM appears as low-signal intensity and OM appears as moderate-signal intensity on T2-weighted images of MRI [50]. The IM is enriched with ER and PR and increases in thickness from the early proliferative phase to the late secretory phase of the menstrual cycle [53,54]. Therefore, it can be presumed that IM may play an important role in female reproduction under the influence of steroid hormones. In contrast, cyclic changes in the expressions of ER and PR in OM are not as robust as in IM. As a major contractile tissue, OM contributes to expulsion of the fetus under regulation of oxytocin and steroid hormones [54,55,56]. There is variable difference in the amplitude and frequency of IM contractions throughout the menstrual cycle. While retrograde contractions of IM (cervix to fundus) during the follicular phase can facilitate sperm transport and promote pregnancy [57], antegrade IM contractions (fundus to cervix) and increased amplitude of peristalsis during menstrual period can promote desquamation of shed endometrium [58,59]. Therefore, a combined effect of repeated episodes of menstruation and associated uterine contractions may disrupt endometrial-myometrial junction. This view supports the association of repeated tissue injury and repair as a mechanism for adenomyosis [60].

## 4. Etiology and Pathogenesis of Adenomyosis

The exact etiology of adenomyosis is currently unclear. Several theories for its development have been proposed, including direct invasion of basalis endometrial cells deep into the myometrium as a result of activation of tissue injury and repair (TIAR) mechanism, metaplasia of displaced embryonic pluripotent Müllerian remnants, and differentiation of embryonic stem/progenitor cells [6,14,15,16]. Since the presence of intravascular adenomyotic tissue was reported in the myometrium of adenomyosis patients, a vascular network may contribute to local migration and spread of these lesions across the uterine wall [61]. Although adenomyosis and endometriosis share some histological features and molecular changes, there are notable differences in their pathogenesis, lesion localization, and clinical features [12,14,61]. Supporting this notion, Koninckx et al. [62] hypothesized in a recent review that a common molecular mechanism might be involved in the pathogenesis of these disorders. However, our knowledge on this hypothesis and on factors that might be responsible for the migration and invasion of basalis endometrium into the myometrium is insufficient. Here we briefly review our current understanding on the etiology and pathogenesis of adenomyosis.

### 4.1. Invagination: Role of Estrogen, TIAR and EMID

Local estrogen production in the eutopic and ectopic endometrium of patients with adenomyosis may play a central role in the etiology of adenomyosis. A state of hyperestrogenism, resulting from increased local aromatization and decreased local estrogen metabolism in the eutopic and ectopic endometria of patients with adenomyosis, has been reported [63]. On the other hand, the activity of aromatase cytochrome P450, a heme-containing enzyme that catalyzes reactions in steroidogenesis is not present in the endometrium of disease-free uteri. An increased estrogen biosynthesis and higher estrogen bioavailability occurs in the eutopic endometrium of patients with adenomyosis, due to local aromatization of circulating androgens into estradiol [63]. The resulting local tissue pooling of estrogen, further supported by reduced conversion of estradiol to less potent estrone, resulting from inactivation of 17β-hydroxysteroid dehydrogenase type 2 (HSD17β2) gene, accelerates cellular proliferation in the eutopic endometrium [64]. While P counteracts estrogen-promoted proliferation in healthy endometrium, this effect is lost or decreased on the endometrium of adenomyosis patients, due to lower immunoreactivity of PR-B secondary to promoter hypermethylation of PR-B, ultimately resulting in P resistance and this aids in fostering abnormal endometrial proliferation [65,66].

In addition to cellular proliferation, estrogen-mediated and oxytocin-promoted uterine activity may result in increased mechanical stress/strains that could injure cells in the junctional zone close to the endo-cornual raphe [60,67,68]. Therefore, an altered endometrial proliferation and hyperperistalsis-induced tissue microtrauma at the EMI may enhance endometrial intramyometrial invagination. In addition, the TIAR mechanism can be activated in response to tissue auto-traumatization [60,67]. Once activated, the TIAR system increases inflammation and local production of estrogen, which, in turn, induce more inflammation and more estrogen production, establishing a feed-forward loop that further induces hyperperistalsis through ER-alpha induction of OT/OTR system [14]. The chronic hyperperistalsis at the JZ facilitates repetitive auto-traumatization, causing disruption of the muscular fibers in JZ, eventually leading to invagination of the endometrial basal layer into the myometrium, which may result in the development of adenomyosis. This biological event at the EMI was recently named as endometrial-myometrial interface disruption (EMID) [14]. It is possible that the TIAR and EMID processes are interconnected. While TIAR can explain the genesis of adenomyosis induced by iatrogenic trauma, it may also explain the pathogenesis of adenomyosis, due to persistent hyperperistalsis that causes EMID. The invagination theory may be consistent with the epidemiological findings that multiparity and uterine surgery, such as D and C are risk factors for adenomyosis, due to their potential to disrupt the JZ. 

A line of evidence demonstrated that the healing process after cyclic tissue injury at EMI can be mediated by the occurrence of tissue hypoxia. Repetitive tissue injury causes disruption of local vasculature and extravasation of blood, leading to platelet aggregation and the formation of clots. Vascular damage results in the loss of perfusion and consequent hypoxia. Once hypoxic, hypoxia-inducible factor-1alpha (HIF-1α) is activated, a key mediator of cellular adaptation to low oxygen levels. In response to hypoxia, macrophages recruited to the wounding site, secrete chemotactic factors and several growth factors (VEGF, PDGF, FGF, TGF-β, HGF) vital for cell migration and proliferation that facilitate tissue repair [14,69,70,71,72]. This function can be aided by activated platelets that release an array of cellular growth factors and angiogenic factors, such as PDGF and VEGF, as well as inflammatory mediators, such as IL-8 [73,74,75]. Collectively, when EMID occurs, the resultant tissue hypoxia would lead to platelet aggregation, increased estrogen production, induction of angiogenic factors, overexpression of COX-2 and PGE2, all of which may promote cell proliferation, mobility and consequent invagination of endometrial cells of the basalis layer into the myometrium.

### 4.2. Role of EMT in Invagination

Now one question remains to be addressed: how do endometrial cells in the basalis layer invaginate deep into the myometrium? A cascade of endometrial-myometrial transition (EMT) at EMI, in response to estrogen and/or inflammation or in response to tissue stress reaction by repetitive TIAR, can be considered. The concepts of epithelial–mesenchymal transition (EMT) and its converse, mesenchymal-epithelial transition (MET) was defined by Elizabeth Hay some 50 years ago [76]. A competent response to EMT-inducing signals can promote disruption of the intercellular adhesion complexes and loss of the apico-basal polarity of the epithelial cells. This feature of EMT is crucial for cells to leave the epithelium and directs migration potentiality. Therefore, events of EMT might occur at EMI, because basalis endometrium and inner myometrium (junctional zone) are closely apposed here without any intervening basement membrane. 

Epithelial-cadherin (E-cadherin) is a multigene family of transmembrane glycoproteins and normally acts as a “cell adhesive zipper”. This means E-cadherin maintains tight cell-cell or cell-matrix contact for epithelial cells [77]. The functional loss in the expression of cell-cell adhesion marker, E-cadherin in epithelial cells and concomitant increased expression of the mesenchymal cell markers, N-cadherin and/or Vimentin, are considered as hallmarks of EMT [15]. It has been demonstrated that epithelial cells, potentially invasive in human endometriosis, lack the expression of E-cadherin [78]. Studies with breast cancer, bladder cancer and hepatoma cells indicated that over-expression of hepatocyte growth factor (HGF) may down-regulate cadherin-mediated cell-cell adhesion with consequent migration and invasion into mesenchymal cells [79]. A line of evidence demonstrated that estrogen and estromedin (estrogen-regulated) growth factor, such as HGF-induced EMT of epithelial cells, may be involved in the development of adenomyosis or may contribute to the metastatic potential of cancer cells [16,80,81].

Previous reports indicated that endometriosis induces a variable amount of stress reaction during cyclic menstrual reflux, implantation and invasion of endometrial cells in pelvis [82,83,84,85]. Similarly, a repetitive TIAR at the EMI in women with adenomyosis may also exhibit tissue stress reaction and tissue damage in the endometrium or in inner myometrium, resulting in the occurrence of inflammation and increased local production of estrogen [14]. In fact, tissue stress reaction may occur in any condition, including endometriosis and adenomyosis, in response to several factors, such as inflammation, physical stress (cell proliferation, invasion), chemical stress (receptor/ligand binding), neurogenic stress, pain sensation, and oxidative stress [86,87]. As a marker of tissue stress reaction, a stronger immuno-expression of HSP70 was observed in the basalis endometrium, as well as inner myometrium in women with adenomyosis, than in control women [86]. This result was corroborated with a relationship between expression of HSP70 and production of different inflammatory mediators and growth factors, including HGF [88,89].

Although the role of EMT in cell migration and dissemination has been reported, our knowledge on the involvement of different cascades of EMT in adenomyosis is poor. The cascade of EMT is composed of three components: (i) disruption of cell-cell adhesion of epithelial origin, (ii) cell migration/invasion, and (iii) transition of epithelial cell phenotype into fibroblast-like phenotype. Khan et al. [16] demonstrated a possible involvement of different cellular cascades of HGF-induced EMT in the development of adenomyosis. This effect of HGF-induced EMT was further promoted by estrogen. The authors found HGF down-regulates gene expression of *E-cadherin* and up-regulation of *N-cadherin* (mesenchymal cell marker) in endometrial epithelial cells (EECs) with an inverse relationship in protein expression between HGF and E-cadherin in gland cells of the basalis endometrium derived from women with diffuse adenomyosis and the ipsilateral side of focal adenomyosis [16]. Moreover, single and combined treatment with HGF and estrogen (E2) demonstrated that ER-positive Ishikawa endometrial epithelial cells expressed higher levels of Slug/Snail, two transcription factors that control EMT by repressing E-cadherin [16]. These findings of HGF up-regulation and down-regulation of E-cadherin with consequent disruption of cell-cell contact of EECs and migration of EECs may be the basic components of EMT in adenomyosis. A separate report demonstrated that the invasion capacity of EECs coincides with migration, taking into account that invasion also depends on migration of EECs. In fact, less expression of E-cadherin in response to HGF and estrogen (E2) corresponded well with migration of EECs [90]. 

Alternatively, platelet aggregation and activation in response to TIAR were also reported to be a possible cause of EMT induction in adenomyosis. This was supported by the fact that platelet aggregation was correlated with the induction of EMT through activation of the TGF-β1/Smad3 signaling pathway in human adenomyosis [91]. We presume that constant tissue stress insult, or an inflammatory response at the endo-myometrial junction in response to repeated D and C or after multiple pregnancies, reported as high-risk factors for adenomyosis, may trigger the very early switch of EMT by higher expression of HGF and lower expression of E-cadherin in the basalis endometrium. Interestingly, the inverse relationship between HGF and E-cadherin was lost once adenomyotic lesion developed [16]. The hyperplastic or hypertrophic changes of surrounding smooth muscle cells and consequent realignment of migrating endometrial cells into the glandular structure may clarify this finding.

A panel of literature has demonstrated lower tissue expression of E-cadherin and increased tissue expression of Vimentin or N-cadherin as dictating a transitional cascade from an epithelial cell phenotype to mesenchymal cell phenotype [77,80,92]. On the other hand, a phenomenon of EMT that was reported to be involved in the metastatic invasion of cancer cells [79,81] might also work in the pathogenesis of adenomyosis. Analysis of the third component of EMT revealed that HGF, either alone or in combination with E2, was able to induce morphological changes of EECs from cobblestone appearance to spindle-shaped fibroblast-like cells. The functional specificity of HGF and E2 was confirmed by their neutralizing effect on EEC motility. These findings give us further information that in addition to functioning as estromedin growth factor (regulated by E2) [16,93] or as pleiotropic growth factor [94] in endometriosis, HGF also exhibits its potential capacity to induce EMT in adenomyosis. Lack of basement membrane in between the basalis endometrium and inner myometrium may better explain EMT at EMI in response to tissue stress insult and/or local inflammation and higher tissue content of HGF and local estrogen in and around EMI. A cascade of EMT in endometrial epithelial cells in response to HGF and estrogen is shown in Figure 2.

### 4.3. Metaplasia of Müllerian Remnants

In addition to the widely accepted invagination theory, adenomyotic lesions may originate de novo from metaplasia of displaced embryonic pluripotent Müllerian remnants. The Müllerian ducts are primordial embryological structures that, during fetal life, develop to form female uterine tract (uterus, fallopian tubes and upper part of the vagina) [95]. These ducts consist of surface epithelium and the underlying urogenital ridge mesenchyme, with the capacity to differentiate into endometrial glands and stroma [96]. It is presumed that metaplastic changes of intramyometrial embryonic pluripotent Müllerian remnants in the adult uterine wall could possibly lead to establishment of de novo ectopic endometrial tissue within the myometrium, resulting in the generation of adenomyotic lesions [2]. The non-infiltrating type of deep endometriotic nodules, which are different from peritoneal endometriosis, may be the consequence of Müllerian rest differentiation, at least in some cases, or the result of an adenomyotic tumoral process originating from the cervix [97,98]. Furthermore, the histological findings of deep endometriotic nodules show identical features of adenomyosis in relation to smooth muscle hyperplasia and fibrosis [97,99]. The Müllerian metaplasia theory is further supported by case reports of confirmed adenomyosis in the rudimentary uterine wall of patients with Rokitansky-Kuster-Hauser syndrome [100,101].

### 4.4. Role of Endometrial Stem/Progenitor Cells 

We cannot rule out the possibility of other de novo development of adenomyosis in addition to invagination theory. Transfer of endometrial stem/progenitor cells to the uterine wall during menstruation and their differentiation into endometrial gland and stromal cells could be another attractive hypothesis. Endometrial epithelial progenitor and mesenchymal stem cells (eMSCs) have been identified in the endometrium and can differentiate into the endometrial epithelial cells and stromal fibroblast-like cells [102,103,104]. Studies with hysterectomy specimens indicated that small populations of both epithelial and stromal stem cells with clonogenic activity were identified in colony-forming units of endometrial epithelial and stromal cells [105]. It is speculated that these stem cells may reside in the endometrial basalis within cell niches [106] and may contribute to cyclic repair of the endometrial functionalis after menstruation [15,107,108]. A study with healthy endometrium indicated that presence of stem cells in the endometrial basalis is a critical factor for regeneration after menstruation and these cells can reach beyond the endometrium [109]. Adult progenitor stem cells may be deposited in the uterus after retrograde menstruation and differentiate into endometrial glands and stroma resulting in de novo development of intramyometrial endometrial implants [15,107,108,109]. An elegant study by Gargett demonstrated that abnormal differentiation of eMSCs can also promote myometrial smooth muscle cells (SMCs) hyperplasia [110]. Therefore, endometrial progenitor and eMSCs may cause adenomyosis in bidirectional pathways, locally via EMI or distal transfer with menstrual debris and its deposition on the uterine wall. 

Several mechanisms have been suggested to clarify the dislocation of endometrial adult stem cells into the myometrium. Consistent with the phenomenon of TIAR at EMI, adult stem cells residing in the basalis layer may cross the EMI, proliferate and differentiate into adenomyotic lesions and promote myometrial SMCs hyperplasia and hypertrophy that are pathognomonic of the disease. In fact, endometrial stem cells may be activated after tissue injury and resulting tissue microtrauma at the JZ and endometrial basalis may trigger establishment of adenomyosis [110]. Moreover, this migration process may be enhanced by elevated expressions of matrix metalloproteinase (MMP)-2 and MMP-9 in endometrial stromal fibroblasts, as a result of enhanced expression of p21-activated kinase 4. An impairment in the expression of p21-activated kinase 4 may disrupt migration of stem cells. Increased expression of these digesting enzymes (MMP-2/MMP-9) may disrupt the normal extracellular matrices and facilitate migration of stem cells into the myometrium [111,112]. Coinciding with Sampson’s theory of retrograde menstruation, the dysregulation of endometrial stem cells has been proposed as a potential mechanism for seeding ectopic endometriotic lesions [113,114,115]. Women with endometriosis shed more basalis than women without the condition during menstruation, thereby providing a means for tissue enriched with endometrial stem cells to migrate to the ectopic sites [113,114]. The same dysregulation process of stem cells may be involved in the development of adenomyosis.

Regarding generation of adenomyotic foci, an alternative hypothesis involves stem-like “pale cells” in basalis endometrium that actively migrate and possibly transit through EMI into the myometrium [116]. At the current moment, it is unclear whether endometrial fragments are shed from the basalis during menstruation in women with adenomyosis like endometriosis, where increased rates of basalis shedding were reported compared with disease-free women [115]. Therefore, the potential role of retrograde menstruation in endometrial stem cell implantation in the uterine wall is still unclear and warrants further study.

### 4.5. Genetic and Epigenetic Alteration in Adenomyosis

Dependence of adenomyosis on ovarian steroids and ovulatory cycles for severity status is a unique feature of the disease. Supporting this context, common patterns of aberrant gene expression have been reported, both in adenomyosis and endometriosis. These include pathways that favor increased estrogen production, decreased estrogen metabolism, ER-β-driven inflammatory process, and progesterone resistance due to decreased PR expression. An epithelial deficiency of the enzyme HSD17β2 causes aromatase overexpression in endometriotic stromal cells. This, in turn, inactivates estradiol by converting it into estrone in adenomyosis [117]. It is likely that HSD17β2 may also be deficient in adenomyotic epithelial cells, as endometriosis and adenomyosis share multiple molecular features. The combination of estrogen excess and HSD17β2 deficiency may give rise to excessive levels of local estradiol in adenomyosis [118,119]. ER-β expression was higher in functionalis glands/basalis stroma and myometrium collected from adenomyotic uteri comparing to ER-alpha gene expression in endometrial glands/stroma [65]. A similar expression of the PR-A and PR-B gene in functionalis endometria was observed with reduced expression in the basalis endometrial stroma and inner/outer myometrium in the adenomyotic uteri [65]. However, in adenomyotic foci, the pattern of ER-β, PR-A and PR-B expression was found to be similar to that in basalis endometrium [65]. Beside hypermethylation of PR, the inverted ratio of ER-α to ER-β may contribute to decreased PR expression [120,121]. In addition, PR-B was suppressed by DNA methylation in adenomyotic stromal cells [66]. Taken together, adenomyotic tissue appears to exhibit progesterone resistance and aberrant estrogen action regulated by ER-β.

Genetic variants involve the key processes in the development of endometriosis. A line of evidence found that cytochrome P450 (*CYP*) genes and catechol-O-methyltransferase (*COMT*) gene variants could influence enzyme activity and increase the risk of estrogen-dependent diseases, such as adenomyosis [122]. Patients with adenomyosis have increased frequency of the C allele in the T/C and C/C genotypes of the *CYP1A1* gene, A allele in the C/A and A/A genotypes of the *CYP1A2* gene, and the T allele in the C/T and C/C genotypes of the *CYP19* gene compared with women without adenomyosis [122,123]. Moreover, *COMT* 158 G/A gene polymorphisms contribute to the high risk of adenomyosis, particularly in Asian populations, further supporting the role of polymorphisms of estrogen metabolism genes in human adenomyosis [122,123,124].

A line of evidence indicated that adenomyosis, like endometriosis, may be considered as an epigenetic disease because epigenetic alterations have been detected in adenomyosis [123,125]. A family of enzymes, such as deoxyribonucleic acid methytransferases (DNMTs), catalyze transfer of a methyl group to DNA, resulting in more compact chromatin and hence gene repression [44,45]. Increased expression of DNMT1 and DNMT3B was found in ectopic endometrium from patients with adenomyosis compared with controls [45]. The mechanism of P resistance with silencing of PR-B, as detected in stromal cells of adenomyotic nodules, can be explained by hypermethylation of the promote region of PR-B [66]. The targeted gene expression to control cellular proliferation, differentiation, and metabolism associated with the development of adenomyosis was associated with DNA hypomethylation and increased expression of a transcription factor, CCAAT/enhancer-binding protein β (CEBPB) [126].

Besides DNA methylation, aberrant expression and localization of class I histone deacetylases (HDACs) was demonstrated in the endometrium of women with adenomyosis. In fact, aberrant expression of histone deacetylases (HDACs) was also found in women with adenomyosis. Indeed, immunoreactivity of HDAC1 and HDAC3 was elevated in eutopic and ectopic endometrium of adenomyosis patients compared with controls [126,127]. These findings were confirmed by the alleviation of dysmenorrhea, hyperalgesia and retardation of myometrial infiltration in patients with adenomyosis after treatment with valproic acid, an HDAC inhibitor [127]. This suggests the involvement of histone modification in the pathogenesis of adenomyosis and confirms the opinion that adenomyosis, like endometriosis, may be an epigenetic disease.

### 4.6. Somatic Mutation in Adenomyosis

Recently good scientific evidence has become available for the cellular origins of adenomyosis by a series of next generation sequencing (NGS) studies that improved our knowledge on the new signaling pathways contributing to its pathogenesis. NGS technologies form the basis of nearly all contemporary genomic approaches, including whole-exome (WES) sequencing and whole-genome (WGS) sequencing. Starting with the widely available use of NGS in the early 2000s, it has been applied to a number of tumors or diseased tissues, including uterine disorders, such as adenomyosis [5]. With the advent of WES and WGS, our understanding of somatic mutations and their involvement in the pathogenesis of different uterine disorders, such as endometriosis, adenomyosis and uterine leimyomas, has greatly improved [5,128,129,130]. In the majority of uterine leiomyomas, enriched smooth muscle cells harbor an identical mutation because of the overwhelmingly high recurrent nature of mutations in a single gene (*MED12*) in the entire tumor [131,132]. In contrast, endometriosis and adenomyosis seemed to originate from the ectopic proliferation and expansion of multiple mutated epithelial cell clones that also contain an attached stromal cell population [129,130,133]. 

In the endometrium, endometriosis, adenomyosis and leiomyoma, the most frequently and activating-type genetic alterations were detected in *PIC3CA*, *KRAS*, *PPP2RIA* and *MED12*, whereas *ARIDIA* mutations lead to a loss of function [128,129,130]. Although genetic alterations accounting for all of the mutations were identified in whole tissues of eutopic endometrium, endometriosis and adenomyosis, the abundantly present stromal cells were mutation-free [128,129,130,133]. Mutational epithelial clones localized in the eutopic endometrial glands seem to play important roles in the pathophysiology of adenomyosis. A number of recurrent driver mutations were found in these endometrial glandular epithelial cells, including mutations affecting *PIK3CA* and *KRAS*. Somatic *KRAS* mutation is a critical genomic alteration associated with adenomyosis [129,130]. The fact that adenomyosis originates from the basalis endometrium is based on the targeted deep sequencing analysis of epithelial cells in adenomyosis and adjacent basalis endometrial glands that demonstrates recurring *KRAS* mutations in both cell types [5,130]. Epithelial cells of the endometrium, adjacent adenomyosis and co-occurring endometriosis also share *KRAS* mutations [129,130]. Mutations found in adenomyosis almost exclusively affected the *KRAS* gene and similar *KRAS* mutated gene was frequently identified in the epithelial cells of endometriotic lesions, whereas *PIK3CA* is the most mutated gene in eutopic endometrial cells [5,130]. In fact, *KRAS* mutations were commonly observed in the adenomyosis (55.6%) and endometriosis (50%) among micro-dissected eutopic endometrial samples, but less frequently in the disease-free group (29.1%) [130]. These findings suggest both adenomyosis and endometriosis are oligoclonal tissues that arise from the endometrial cell populations carrying a specific driver mutation most commonly affecting the *KRAS* gene. Based on these NGS data, it is tempting to assume that once trapped within the myometrium during traumatic processes at the endometrial-myometrial junction, adenomyosis primarily stems from the basalis portions of the endometrial glands that harbor a *KRAS* mutation.

### 4.7. Role of microRNAs in Adenomyosis

A growing body of evidence shows that the remodeling of retrograde endometrial tissues to the ectopic endometriotic lesions involved multiple epigenetic alterations, such as DNA methylation, histone modification, and microRNA (miRNA) expression [134]. Despite abundant publications on the involvement of miRNAs in endometriosis, our understanding on the role of miRNAs in adenomyosis is unclear. MicroRNAs belong to the group of single-stranded, non-coding RNAs with an average size of 22 nucleotides. They play important regulatory roles in gene expression through pairing with messenger RNA (mRNA) to modulate RNA splicing, degradation, and translation [135]. The genome-wide analysis of the miRNA expression profile demonstrated that dysregulated miRNAs play critical roles during the development of endometriosis through modulating the cell cycle progression, apoptosis, steroidogenic pathway, hormone signaling, inflammation, and response to hypoxia [136,137]. 

The different cellular cascades of EMT are involved in the development of adenomyosis, as already mentioned above. It has been reported that epithelial-mesenchymal phenotypic plasticity is regulated by a set of regulatory feedback loops involving TGFβ signaling, members of miRNA-200 family and variable transcription factors, such as zinc finger E-box (ZEB)1 and ZEB2 [138]. When TGFβ signaling is up-regulated, a system shift towards low miRNA-200 expression and high ZEB1/ZEB2-mediated E-cadherin suppression may promote a mesenchymal phenotype indicating that similar signaling pathways may be involved in the pathogenesis of adenomyosis. Similarly, metastatic spread of tumor cells is also promoted by miRNA regulated EMT [139]. It has been reported that transcripts of miRNA-20a, a member of the miRNA17-92 cluster, miRNA-22, and miRNA-26a were downregulated in ectopic endometrial lesions compared to eutopic endometrium [140]. On the basis of these findings, it can be presumed that the collective dysregulation of a variable miRNAs, such as miRNA-16, miRNA-17, miRNA-21, miRNA-22, and miRNA-26a, strictly controls cell cycle and may be involved in the progression of endometriosis and other reproductive diseases, such as adenomyosis [140]. Consistent with these findings, a separate study indicated that expression of miRNA-17 was increased in the endometrial tissues of patients with adenomyosis and might influence cell apoptosis and cyclin expression through the targeted down-regulation of the gene phosphatase and tensin homolog (PTEN). These findings suggest that miRNA-17 may promote the occurrence and/or development of adenomyosis [141].

Let-7A is a small non-coding RNA that has been found to take part in cell proliferation and apoptosis. The hippo-YAP1 axis, known as the tumor suppressor pathway, also plays an important role in cell proliferation and apoptosis [142,143]. A recent report found that Let-7A level was decreased, while the YAP1 level was increased, in uterine junctional zone SMCs in adenomyosis. Up-regulated Let-7A affected the expression levels of the components of the hippo-YAP1 axis, accelerated apoptosis and inhibited proliferation of JZ SMCs [144]. These findings suggest that the Let-7A and hippo-YAP1 axis may act as important regulators in the proliferation of JZ SMCs and occurrence of adenomyosis. Several circulating miRNAs are differentially expressed in the sera of patients with endometriosis. While a combination of serum levels of Let-7A-7C during the proliferative phase have been proposed to serve as a diagnostic marker of endometriosis [145], the diagnostic role of these small non-coding RNAs in women with adenomyosis is yet to be determined.

## 5. Clinical Classification of Adenomyosis

Clinically, adenomyosis may present in different configurations in the myometrium, such as focal, diffuse and rare cases of cystic adenomyoma, and is better detected by magnetic resonance image (MRI) [1,146,147] than other imaging modalities. Focal adenomyosis was considered when circumscribed nodular aggregates were observed on either the anterior or posterior wall of the uterus, and diffuse adenomyosis when numerous foci of endometrial glands and stroma were dispersed diffusely within the myometrium [148]. Diffuse adenomyosis was defined by the following criteria on MRI: (i) maximum thickness of the Junctional Zone (JZ_max_) of at least 12 mm as a result of hyperplastic change or distortion of JZ causing scattered invasion of basalis glands into the myometrium [8,17], (ii) JZ_max_ to myometrial thickness ratio of >40% [149].

It has been claimed that classical hypothesis of adenomyosis supporting its origin from basalis endometrium is not always justified. There are cases in which no direct relationship between adenomyosis and the endometrium can be histologically proved; rather the disease appears to be the result of the invasion of endometrium-like structures from outside of the uterus that disrupts the uterine serosa [150]. Based on their clinical experience, and assessed by MRI and histology, the authors proposed four subtypes of adenomyosis: (i) subtype I or intrinsic adenomyosis was defined by a product of direct endometrial invasion involving inner-mid myometrium without affecting the outer structures, (ii) subtype II, or extrinsic adenomyosis, consisted of adenomyosis that occurs in the outer uterine layer without affecting the inner structures, (iii) subtype III, or intramural type, consisted of adenomyosis that occurs solitarily without relationship to structural components, (iv) subtype IV, or indeterminate type, was defined by adenomyosis that did not satisfy other criteria [150]. 

Based on the surgical and histologic profiles of this study, while patients with intrinsic adenomyosis had 6.8–25.4% detection rate of deep infiltrating endometriosis (DIE), patients with extrinsic adenomyosis displayed a significantly higher detection rate (92.3–96.1%) of coexistent lesions of DIE [150]. Two more recent studies [6,151] reported that focal adenomyosis located in the outer myometrium was observed more frequently in women with endometriosis and was significantly associated with the DIE phenotype, supporting the extrinsic type of adenomyosis, as proposed by Kishi et al. [150]. Regarding nomenclature of adenomyosis, Bazot and Darai [152] already proposed internal and external adenomyosis. The new nomenclature as proposed by Kishi et al. [150], intrinsic and extrinsic adenomyosis, may be synonymous terminology to adenomyosis interna and adenomyosis externa, respectively. It is still unclear, and a debatable issue, whether we should consider extrinsic adenomyosis as a variant of adenomyosis or whether it is truly a subtype of endometriosis originating in the pelvis, which subsequently invaginates into the sub-serosal area of the outer myometrium during the progressive course of the disease. 

### 5.1. Biological Differences between Focal and Diffuse Adenomyosis

Medical therapies for women with adenomyosis include suppressive hormonal treatments, such as continuous use of oral contraceptive pills, high-dose progestins, selective estrogen receptor modulators, selective progesterone receptor modulators, the levonorgestrel-releasing intrauterine device, aromatase inhibitors, danazol and gonadotropin-releasing hormone agonist (GnRHa) [153,154]. All these hormonal medications can temporarily induce regression of adenomyosis and improve the symptoms. However, none of them have been clearly validated for therapeutic effect in adenomyosis. A time-dependent recurrence of adenomyotic lesions and/or symptoms is a common feature in women with adenomyosis as a result of either less efficacy of treatment or resistance to hormonal medications. Depending on the individual patient, resolution of symptoms appears to be a more common finding than regression of uterine size after hormonal treatments [155]. Our understanding on the biological differences between focal and diffuse adenomyosis is unclear and information is scant to clarify the mechanistic basis of hormonal resistance to adenomyosis.

Immunohistochemical analysis using tissues from focal and diffuse adenomyosis that were collected during hysterectomy revealed that there was no difference in ER and PR expression in gland cells or stromal cells of adenomyosis lesions on the ipsilateral side of focal adenomyosis and the anterior/posterior walls of diffuse adenomyosis [156]. Compared to myoma tissues, a relatively decreased expression of ovarian steroid receptors was observed in both focal and diffuse adenomyosis [156]. These findings of substantially lower ER and PR expressions in adenomyotic foci, myometrium and vasculatures are further pieces of evidence that a proportion of women with focal and diffuse adenomyosis may be resistant to estrogen suppressing agents or progestational compounds improving their clinical symptoms or decreasing uterine size. A failure to decrease in ER and PR expression in adenomyotic foci and myometrium in response to GnRHa may support this notion for women with diffuse adenomyosis. In contrast, women with focal adenomyosis may be responsive to estrogen-suppressing agents while their responses to progestin are questionable. A separate study indicated that *KRAS* mutations are more frequent in cases of adenomyosis with co-occurring endometriosis, low PR expression or progestin pretreatment [130]. This study confirmed the diminished anti-proliferative effect of progestin compound via epigenetic silencing of *PR* in immortalized cells with *KRAS* mutation [130].

Rigidity of the uterus is another important factor that may also explain the poor responsiveness to hormonal medications in women with adenomyosis. A variable amount of tissue inflammatory reaction in the myometrium, with consequent accumulation of collagen fibers, collagen matrix and fibrous elements, may time-dependently cause fibrosis and stiffness of the myometrial tissue. Image analysis of tissue fibrosis indicated that amount of fibrosis was stronger in both focal and diffuse adenomyosis, compared to fibrosis in the myometrium derived from women with uterine myoma. The pattern of fibrosis was not different in tissues derived from GnRHa-treated and -untreated women with focal and diffuse adenomyosis. Collectively, the expression profiles of ER/PR and entity of fibrosis were not different between women with focal and diffuse adenomyosis regardless of GnRHa treatment and may clarify the biological rationale of non-response to hormonal therapies for adenomyosis. Aside from fibrosis-related hormonal resistance, the most recent study by Hunag et al. [157] demonstrated that extension of fibrosis in adenomyosis lesions and its propagation to neighboring EMI and eutopic endometrium contributed to heavy menstrual bleeding via reduction of PGE2 and HIF-1α signaling. A critical question now remains to be addressed. Why does GnRHa improve pregnancy rate in ART treatment even though has no effect on ovarian steroid receptor expression or in decreasing fibrosis or uterine size? The effect of GnRHa on increasing implantation and pregnancy rates after ART in women with adenomyosis may be explained by decreasing inflammatory reaction in the endometria, prostaglandin production with consequent decrease of hyperperistalsis, and improvement in sperm/embryo migration or resolving uterine auto-traumatization [158,159,160]. 

### 5.2. Biological Differences between Intrinsic and Extrinsic Adenomyosis

There is no doubt that the subtype classification of Kishi et al. [150] is important for all surgeons in order to do surgical planning to remove each type of adenomyosis, including intrinsic and extrinsic adenomyosis. However, from the pathological and physio-pathological perspectives, extrinsic adenomyosis could arise from outside of the uterus, such as endometriosis, but any biological evidence related to this issue is lacking. We came to learn from a recent study that the morphological appearance of tall columnar cells with abundant stromal cells around glands in intrinsic adenomyosis were almost similar to the endometrial gland cells and stroma, supporting their origin from the endometrium. In contrast, the pattern of Ber-EP4-immunoreacitve gland cells were mostly thin and amounts of CD10-positive stromal cells were scanty in extrinsic adenomyosis that were closely similar to their coexisting DIE lesions [151]. These findings may explain their possible origin from extra-uterine sources, such as DIE.

Immunohistochemical analysis of ovarian steroid receptors revealed that while ER/PR expression was not significantly different in glands/stromal cells of extrinsic adenomyosis, PR-immuno-stained cells were significantly lower than ER in the glands/stroma of intrinsic adenomyosis. The ER and PR expression patterns were similar in the glands/stroma of coexistent DIE lesions in these two types of adenomyosis. ER and PR expressions were not different in extrinsic adenomyosis and their coexisting DIE lesions [151]. Relatively more fibrosis was seen in biopsy samples of extrinsic adenomyosis and coexistent DIE than in intrinsic adenomyosis and their coexistent DIE without showing any significant difference between them. In fact, fibrosis in extrinsic adenomyosis and its coexistent DIE almost occupied the rim of stromal cells around the glands in these lesions [151]. This biologically significant finding may give us some hint on the controversial debate as to whether we should consider extrinsic adenomyosis as an “adenomyosis interna” or as an “adenomyosis externa”. These notions may support previously reported proposals that we should define DIE pathologically as either adenofibroma or adenomyosis externa [124,161].

Despite existing debates on this issue, based on recent findings, a potential biological difference between intrinsic and extrinsic adenomyosis cannot be ignored and extrinsic adenomyosis may be considered as adenomyosis externa, due to a close histological and biological relationship between extrinsic adenomyosis and coexistent DIE. The localization of adenomyosis foci on the outer myometrium (extrinsic adenomyosis) may be explained by direct invagination of coexistent DIE lesions into the cervix and their ascending migration/extension along the uterine serosa and, as such, may be considered as adenomyosis externa. A careful search of adenomyosis in the outer myometrium may be necessary in clinical practice for women who harbor DIE, also called deep endometriosis (DE), in the pelvis. One important question still needs to be investigated. Could deep endometriosis be the progenitor of the extrinsic type of adenomyosis? We expect that future study will address this unclear pathogenic process.

## 6. Association with Infertility

Female infertility and subfertility are clinical conditions associated with a significant economic and psychosocial impact [162,163,164]. There are many gynecological diseases that influence infertility including endometriosis [165], ovulatory dysfunction [166], tubal factor [167], endocrine disruption [168], reduced endometrial receptivity [169,170], and age-related infertility [171]. Infertility is highly prevalent among women with endometriosis (25–50%). Its etiology is ambiguous and the exact mechanisms driving infertility are unclear, as the majority of women with endometriosis are able to conceive but with reduced fertility [165,172]. Similarly, the mechanisms causing infertility or subfertility in women with adenomyosis are elusive, because the majority of women with adenomyosis are multiparous. Approximately, 20% of cases of adenomyosis involve women younger than 40 and 80% are aged 40 to 50 years, when they have almost completed their childbearing activity [25]. Recently, however, an association between adenomyosis and infertility has emerged. With the advent of non-invasive diagnoses with MRI and TVUS, the role of adenomyosis in infertility and early pregnancy has been better recognized [173,174].

A potential concern exists in the majority of reported studies to find an association between adenomyosis and infertility as adenomyosis commonly coexists with other pathologic processes linked to infertility, such as endometriosis, polyps or leiomyomas [175]. Endometriosis is reported to occur in 54–90% of cases with adenomyosis [8,176]. Therefore, we cannot avoid the bias that the cause of infertility is due to concurrent endometriosis rather than adenomyosis, because endometriosis is a well-known condition causing infertility [177]. However, a study with baboons showed a strong association between histological adenomyosis and lifelong infertility, even in cases when coexisting endometriosis was excluded [178]. This was confirmed in another study of women who received embryos created through oocyte donation. In this study, the miscarriage rate was significantly higher in groups of women who had adenomyosis alone versus those with co-existing endometriosis or controls [179]. A recent meta-analysis further concluded that adenomyosis has a detrimental effect on clinical outcomes of in vitro fertilization [180].

### 6.1. Proposed Mechanisms

A recent trend is that women delay their first pregnancy until they are aged in their late 30s or early 40s and, as such, adenomyosis has been diagnosed with increasing frequency in infertile women [181]. Although the exact mechanism behind the relationship between adenomyosis and infertility is still unclear, a number of factors has been proposed and focus on four putative pathways: (i) Intrauterine abnormities and increased uterine peristalsis causing abnormal utero-tubal sperm transport. Intrauterine anatomical distortion caused by uterine hyperperistalsis and inflammation-induced adnexal adhesion may block the tubal ostia and potentially impair sperm migration and embryo transport. The abnormal myometrial contraction waves lead to abnormal sperm transport through the uterine cavity and may also lead to intrauterine pressure [181,182,183]. (ii) Abnormal endometrial steroid metabolism, increased inflammatory response, and increased intrauterine oxidative stress environment leading to altered endometrial function and receptivity [169,170]. The increased density of macrophages (Mφ) increases inflammatory response of the endometrium and release of reactive oxygen species that are thought to be embryotoxic [184]. (iii) Impairment of implantation may result from inflammation, a lack of adequate expression of adhesion molecules (integrins), reduced expression of implantation markers, such as leukemia inhibitory factor (LIF), and altered function of the gene for embryonic development (HOXA10) [185]. (iv) Occurrence of chronic endometritis (CE) resulting from intrauterine microbial infection may be associated with negative fertility outcome in women with adenomyosis [186]. 

Recent studies have shown a correlation between CE and reproductive failures, such as recurrent implantation failures (RIF) after IVF-ET, recurrent miscarriage and unexplained infertility [187,188]. The major cause of CE is microbial infection in the uterine cavity. This is supported by the fact that treatment with antibiotics is effective to eliminate plasma cells in the affected patients [188,189,190]. A multicenter cohort study in Japan reported higher incidence of uterine infection in patients with diffuse adenomyosis that may result in the occurrence of chronic endometritis (CE) in these women [191]. Although the causality between CE and embryo implantation failure is controversial, reports suggest that CE negatively affects reproductive outcome. A recent study provides the first piece of clinical evidence that a variable rate of chronic endometritis (CE) occurs in women with different types of adenomyosis, such as focal and diffuse adenomyosis [192]. Although an insignificant difference in the occurrence of CE was found between focal (58.8%) and diffuse adenomyosis (60.0%), the ipsilateral side of focal adenomyosis showed a significantly higher occurrence of CE (58.8%) than the contralateral side (11.7%) [186]. These findings indicated that a variable occurrence of CE in different types of adenomyosis may be involved in negative fertility outcome. 

Similar to endometriosis, where inflammation is a common factor associated with infertility and chronic pelvic pain, a similar inflammatory response of the endometrium may play a significant role in the adverse reproductive outcome in women with adenomyosis [179,180,181,182,183]. In contrast to women with endometriosis, adenomyosis has not yet been shown to have an adverse influence on oocyte function or folliculogenesis [193]. In patients with endometriosis, different inflammatory markers (macrophages, prostaglandins, IL-1, IL-6, TNFα) were increased in the peritoneal fluid and their high concentrations may negatively affect oocyte function [158,194,195]. However, no association has been found so far between adenomyosis and oocyte quality or function.

### 6.2. Role of Microvilli and Axonemal Alteration

A successful spontaneous conception requires normal function of endometrium and Fallopian tube and this contributes to a physiologically optimized environment for fertilization and early embryonic development. This provides a conduit for the gametes to convene and for the embryo to reach the uterine cavity [196]. The successful capture and/or migration of sperm and embryo may be achieved by the efficient microtubule-mediated movement of microvilli in the apical surface of endometrium [160]. Adenomyosis-induced local inflammation is one of the biological bases for a negative impact of adenomyosis on fertility [192]. Negative fertility outcome in women with adenomyosis could be due to tissue inflammation of the endometrium and/or toxic effect of chemical mediators as released by different immune cells [158,194,195]. If these embryotoxic chemical mediators diffuse to the apical endometrial cells, they may cause structural damage to the apical microvilli and its core bundles of microtubules [160]. In fact, inflammation-induced damage of mucosal cilia in the Fallopian tube has been described in women with ectopic pregnancy and salpingitis [196]. These microtubules are similar in arrangement to those in the Fallopian tube. Normal microtubules are arranged in a 9 + 2 pattern, in which nine peripheral microtubule doublets surround a core of two central single microtubules, encased within a single microvillus and designated as normal pattern and arrangement of axoneme [197,198,199,200].

In an attempt to find an association between endometrial inflammation, microvilli damage and an axonemal alteration in the apical endometria, a recent prospective cohort study was performed using endometria derived from women with and without adenomyosis. An in-depth evaluation with transmission electron microscopy (TEM) found that, compared to control endometria, the number of microvilli on the apical epithelial cells of endometria collected from women with focal and diffuse adenomyosis was significantly decreased, resulting in marked abnormality in their axonemal arrangement [160]. While the contralateral side displayed significantly less abnormal microtubules, the ipsilateral side of focal adenomyosis showed significantly higher abnormality, compared to normal patterns of microtubules. These findings were consistent with strong tissue inflammatory reaction in endometria collected from women with focal and diffuse adenomyosis, compared to that of control women with both fibroids and CIN3 [160]. Interestingly, endometria collected from symptomatic women with focal adenomyosis showed significantly increased tissue inflammatory reaction, compared to that of asymptomatic women. These biological and ultra-structural abnormal findings in the endometria may be associated with negative fertility outcome in women with adenomyosis. The distribution of abnormal axonemal arrangements was more frequently observed in women with symptomatic adenomyosis than in asymptomatic women [160] and supported the findings of some ART clinical trials [201,202]. The possible mechanisms that might be involved in infertility or subfertility in women with adenomyosis are shown in Figure 3.

An Italian study demonstrated that clinical pregnancy rate, implantation rate, and live birth rate are not impaired in asymptomatic women with adenomyosis, compared to groups of women without adenomyosis in IVF cycles [201]. On the other hand, a systemic review and meta-analysis targeting IVF outcome suggested that women with symptomatic adenomyosis have a 28% reduction in the likelihood of clinical pregnancy rate (RR = 0.72; 95% CI, 0.55–0.95) and two-fold increase in the risk of miscarriage (RR = 2.12; 95% CI, 1.20–3.75) [202]. These findings indicate that complaints of symptoms may be associated with a causal link between adenomyosis and infertility. In addition to other mechanistic bases, as mentioned above, the ultra-structural abnormalities of microvilli and microtubules in the apical endometria in response to tissue inflammatory reaction may clarify the possible association between negative fertility outcome and adenomyosis. We have several diagnostic tests in infertility clinics, such as hysterosalpingography (HSG) or laparoscopic chromo-pertubation, but these tests are not capable of evaluating endometrial or tubal physiology, except understanding of the state of tubal patency and/or endometrial pathology. In fact, the functional integrity of apical endometrial cells is required for the establishment of a successful pregnancy. Although a direct effect of microvilli and axonemal alteration on infertility is unclear, the findings by Khan et al. [160] may be clinically useful during counseling with symptomatic patients with adenomyosis who are planning to conceive naturally or by ART. 

## 7. Conclusions and Future Perspective

With the elapse of more than one hundred and fifty years since the report of Von Rokitanski in 1860, most of the literature still claims that the pathogenesis and pathophysiology of endometriosis and adenomyosis is unclear. Despite abundant publications, lack of standardized histopathologic criteria for diagnosis and variable numbers of histologic tissue samples evaluated per hysterectomy lag behind exact information on the true incidence rate of adenomyosis.

However, the bulk of recent evidence has improved our knowledge greatly on the pathogenesis of adenomyosis and supports the view that adenomyosis may result from one of several mechanisms: (i) Invasion of the endometrial basalis into the myometrium by enhanced cell survival, EMT and cell migration. Repeated tissue injury and repair at EMI caused by uncontrolled menstruation and EMID acts as initial trigger for EMT. As chemical mediators, estrogen and estromedin (estrogen regulated) growth factors, such as HGF at EMI, may trigger the sequential cellular cascades of EMT. Since, adenomyosis can also occur spontaneously in many animal species, the role of EMID in the pathogenesis of adenomyosis has been recently confirmed in an animal model to strengthen the validity and feasibility of findings in human study [203]; (ii) De nevo metaplasia of displaced pluripotent embryonic Mullerian remnants or differentiation of adult stem cells; (iii) There are also genetic, epigenetic and environmental factors that may trigger the pathogenic process of adenomyosis; (iv) The recently discovered epithelial mutations that mostly affect the *KRAS* gene and are seen in both the glandular endometrium and adjacent adenomyosis provide high quality evidence that adenomyotic gland islands arise from the basalis layer of the endometrium. (v) Multiple epigenetic alterations, such as DNA methylation, histone modification, and differential expression profiles of microRNAs (miRNAs), may be involved in the occurrence of endometriosis and adenomyosis.

With the understanding that the menstruation process itself may increase the risk for the entrapment of fragments of the basalis layer within the myometrium, some questions still remain to be addressed: (i) Why and how does adenomyosis occur in response to *KRAS* mutation and how can oligoclonal expansion of cells in basalis endometrium switch these cells to enter into the myometrium by crossing the EMI? (ii) Is there any association between *KRAS* mutated/surviving cells in basalis endometrium and cellular cascades of EMT in the development of adenomyosis? (iii) If stromal cells are non-mutated and less survived, what is the role of endometrial stromal cells in the occurrence of endometriosis or adenomyosis? In fact, histologically, endometrial glands or glandular epithelial cells in ectopic locations are encased with a variable amount of surrounding stromal cells. We should keep in mind that stromal endometriosis was detected in a proportion of cases during searches of occult microscopic endometriosis in visibly normal peritoneum [204]. Further in-depth studies on these unclear issues may illuminate new directions for better understanding of the exact pathogenesis and pathophysiology of adenomyosis. Although yet to be investigated, and already reported for endometriosis, measurement of single or combined serum levels of some target miRNAs across the phases of the menstrual cycle may provide new diagnostic markers in women with adenomyosis.

The current review has also improved our knowledge and understanding of the biological differences between functionalis and basalis endometrium, between focal and diffuse adenomyosis, as well as between intrinsic and extrinsic adenomyosis, the two recently proposed subtypes of adenomyosis. These findings are important for clinical gynecologists who are involved in medical and surgical management of adenomyosis. The fact that adenomyosis is not simply endometriosis of the uterus implies that, when it comes to management and prevention, careful perioperative interventions should be carried out and monitored for adenomyosis, such as D and C, in order to reduce the risk of adenomyosis. However, there are many unknowns and our knowledge on adenomyosis is still immature. 

It seems reasonable to hypothesize the existence of a causal link between adenomyosis and infertility/subfertility. Most of the studies investigating adenomyosis as a possible cause of infertility have focused on the comparison of clinical outcomes of ART procedures between affected and non-affected infertile women. The rationale for this approach is that it allows evaluation of the influence of adenomyosis on embryo implantation. The biological basis for a negative impact of adenomyosis on fertility may include one of the followings: adenomyosis-induced local inflammation, impairment of utero-tubal sperm transport, altered endometrial function/receptivity, and dysregulation of local hormonal metabolism leading to hyperestrogenic milieu. According to the most recent information, endometrial inflammation-induced microvilli damage and an axonemal alteration in the apical endometria may clarify a link between adenomyosis and negative fertility outcome [160]. These recent findings may be clinically useful during counseling with symptomatic patients with adenomyosis who are planning for future pregnancy. 

Unfortunately, there are several factors that make it difficult to investigate the relationship between adenomyosis and infertility: (i) the incidence of adenomyosis is not correctly known, (ii) universally accepted diagnostic criteria for adenomyosis are still lacking, (iii) adenomyosis often coexists with endometriosis and/or uterine fibroids. There is an unmet need for adequately designed prospective studies in order to improve our knowledge of this polymorphic disease, to consequently establish more effective therapeutic strategies and to evaluate the cause-effect relationship between adenomyosis and infertility. Future research is warranted to elucidate the molecular mechanisms underlying the different pathogenic processes of adenomyosis. Integrating current knowledge as described in this review article and advances in new directions, such as novel models, novel approaches, new technologies, are essential tools to understand the exact pathogenesis of adenomyosis and its association with infertility.

## Figures and Tables

**Figure 1 jcm-11-04057-f001:**
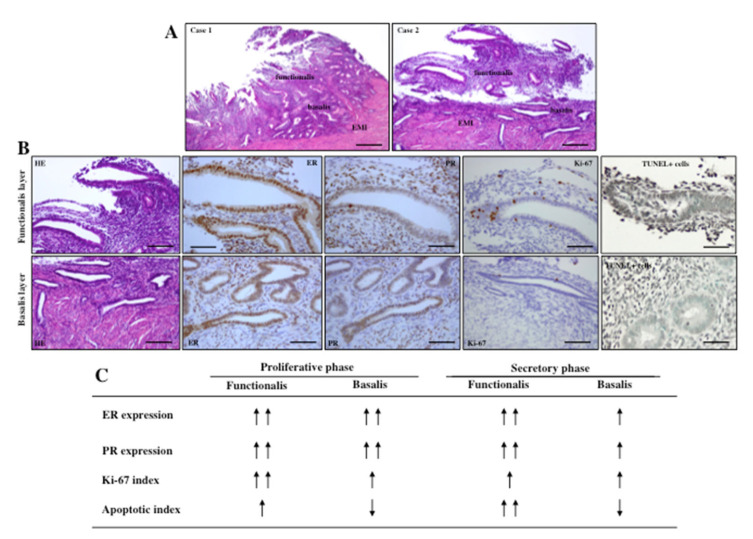
(**A**) Hematoxylin and eosin (HE)-stained slides showing degradation of functionalis endometria and intact basalis endometria derived from the hysterectomy specimens of two different cases with adenomyosis during menstruation. (**B**) Immunohistochemical staining of estrogen receptor (ER), progesterone receptor (PR), Ki-67 (cell proliferation marker), and TdT-mediated dUTP-biotin nick end-labeling (TUNEL)-positive cells in the functionalis endometria (upper row) and basalis endometria (lower row) derived from the same patient with adenomyosis during menstruation. (**C**) Shows the immunohistochemical expression patterns of ER, PR, Ki-67 index, and apoptotic index in the functionalis and basalis endometria during the proliferative phase and secretory phase of the menstrual cycle. These slides were reproduced with permission from the article of Khan et al. [17]. Scale bar = 50 μm or 100 μm.

**Figure 2 jcm-11-04057-f002:**
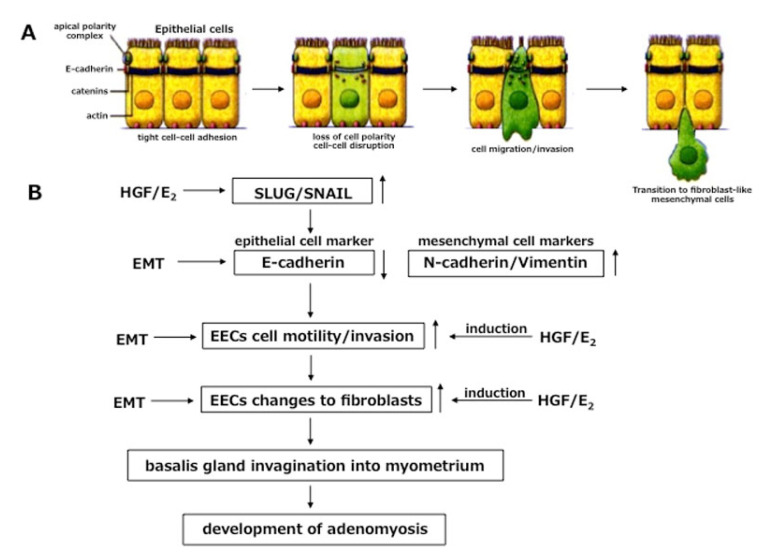
(**A**) Shows cellular aspects of three different components of epithelial-mesenchymal transition (EMT) such as cell-cell disruption, cell migration/invasion and changes of endometrial epithelial cells to spindle-shaped fibroblast-like mesenchymal cells that may be involved in the development of adenomyosis. The revised model of this diagram is extracted from the article of Acloque H et al. [92]. (**B**) A diagrammatic representation showing hepatocyte growth factor (HGF)- and estrogen (E_2_)-induced pathways in the occurrence of epithelial-mesenchymal transition (EMT) in human adenomyosis. Up-regulation of SLUG and SNAIL, two transcriptional repressors of E-cadherin, in response to HGF and E_2_ are associated with decreased expression of E-cadherin (epithelial cell marker) and increased expression of N-cadherin/Vimentin (mesenchymal cell markers) causing disruption of tight cell-cell contact. These cellular events in endometrial cells and in intact tissues trigger morphological changes of endometrial epithelial cells (EECs) to a fibroblast-like mesenchymal phenotype and induce increased cell migration/invasion, two essential components of EMT that induced by HGF and E_2_ either alone or in combination. All these events of EMT may be involved in adenomyosis.

**Figure 3 jcm-11-04057-f003:**
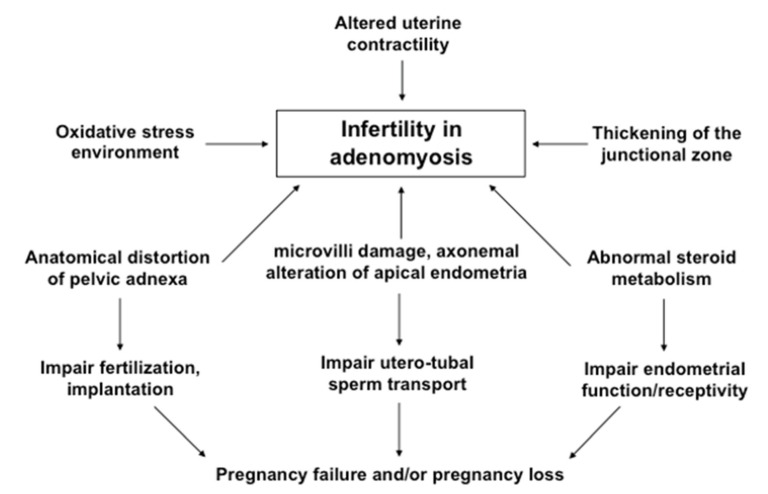
Represents different mechanistic bases that have been proposed and may be involved in the occurrence of negative fertility outcome in women with adenomyosis.

## Data Availability

Not applicable.

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
