# Peer review of "Pathogenesis of Human Adenomyosis: Current Understanding and Its Association with Infertility"

_jcm, 2022, doi:10.3390/jcm11144057_

Round 1

Reviewer 1 Report

 I have accurately read the manuscript  entitled: "Pathogenesis of human adenomyosis: current understanding and its association with infertility"
I need to report some necessary changes because it cannot be published as it stands.
- I suggest a minor English revision performed by a native English speaker.
- Introduction needs to be more concise, without long historical digressions.
- This review lacks of a sistematic structure, because the authors do not mention the most important features in the study selection: the involved years and the databases included. 
- The development of the major body seems to be ambiguous without a clear organization. The authors should avoid the inclusion of personal thoughts, which is not suitable for a review. 
- the aim of this review is not completely understandable and the link between adenomiosis and infertility seems to be poorly discussed. 
- Discussion needs to be adequately improved to strenghten the results proposed in the manuscript
- The authors should provide more recent citations because it cannot published as it stands  

All things considered, I suggest a major revision.

Reviewer 2 Report

Fellow doctors, 

Congratulations on your work! 

Although the article does not bring original novelty research to the field, I think it makes a valuable summary of the existent information on the subject.

I particularly appreciated the chapter regarding endometrial-myometrial transition (EMT), the genetics (NGS and WGS) and the influence of adenomyosis in infertility.

Reviewer 3 Report

The authors discuss the pathophysiology of adenomyosis highlighting the molecular alteration in EMT, estrogen mechanism, stem cell, epigenetic alteration and Mullerian pathways. The MS has attempted to provide a mechanistic connection between infertility and adenomyosis. The pathology of normal endometrium vs adenomyosis was discussed. The expression profile of the markers progesterone receptor, estrogen receptor, ki67 and TUNEL cells (apoptotic index) were provided between the early proliferative and secretary phase of menstrual cycle of adenomyosis patients. The cellular proliferation, inflammation induced by tissue injury and repair and progression through Endometrial- myometrial interface disruption was also discussed in the review. Adenomyosis progression by EMT through the activation of TGF-B and SMAD was discussed, the switch from Epithelial to Mesenchymal Phenotype was also discussed with the differential expression of various markers like E-Cadherin, vimentin and N cadherin.

However, the role of Stem cells in adenomyosis seems incomplete as the molecular dysregulation commonly noted were not explained.

Genetic alteration mentioned are outdated and does not line with the current findings.

Epigenetic alterations mentioned fall short as it does not provide a full picture of the modifications

Role of MicroRNAs needs to be included.

The biological difference between the types of adenomyosis is very well established with both Histological and Genetic criteria and hence photomicrographs & suitable images will added value to the discussion.

The impact of adenomyosis on fertility is discussed with the special emphasis on microvilli and axonemal alterations, but this part is inconsistence and lacks coherence.

  1. incidence rates of Adenomyosis are not well defined, conclusive data with suitable refence is needed
  2. lines 115-120 in page 5 are confusing and inconclusive, needs to be rewritten.
  3. the differentiation of Endometriosis and Adenomyosis is not well defined & hence needs to be articulated
  4. Lines 231-235 needs to be elaborated as to what genes are activated, leading to cellular proliferation
  5. Effect of MicroRNAs on disease progression is not discussed. Micro RNA MiR-17, MiR-22 and Let-7A are important in adenomyosis pathogenesis
  6. The conclusion appears to be redundant & hence needs to be rewritten.

Reviewer 4 Report

It is a well-writen, with updated literature on an interesting toping. Nonetheless, I would recommend to write again the Abstract, since it is not a good reflection of what is the paper about. On the other hand, it is a review article, so I would not include the classification of adenomyosis that your group is proposing, specially because there is no validation nor explanation of this new classification. 

Round 2

Reviewer 1 Report

All the changes that the authors have done improve the quality of this study.

Reviewer 3 Report

This Manuscript with so much Plagiarism (66%) and need to revise highly repetitive sentences.
